# Towards the application of Tc toxins as a universal protein translocation system

Daniel Roderer [1], Evelyn Schubert [1], Oleg Sitsel [1] & Stefan Raunser [1]*

Tc toxins are bacterial protein complexes that inject cytotoxic enzymes into target cells using a syringe-like mechanism. Tc toxins are composed of a membrane translocator and a cocoon that encapsulates a toxic enzyme. The toxic enzyme varies between Tc toxins from different species and is not conserved. Here, we investigate whether the toxic enzyme can be replaced by other small proteins of different origin and properties, namely Cdc42, herpes simplex virus ICP47, *Arabidopsis thaliana* iLOV, *Escherichia coli* DHFR, Ras-binding domain of CRAF kinase, and TEV protease. Using a combination of electron microscopy, X-ray crystallography and in vitro translocation assays, we demonstrate that it is possible to turn Tc toxins into customizable molecular syringes for delivering proteins of interest across membranes. We also infer the guidelines that protein cargos must obey in terms of size, charge, and fold in order to apply Tc toxins as a universal protein translocation system.

[1] Department of Structural Biochemistry, Max Planck Institute of Molecular Physiology, Otto-Hahn-Str. 11, 44227 Dortmund, Germany.
*email: stefan.raunser@mpi-dortmund.mpg.de

Bacteria produce numerous pore-forming toxins that function by puncturing the plasma membrane of target cells. There, they either form a perforating pore that dissipates crucial electrochemical gradients or function as an injection device that translocates a toxic molecule into the cytoplasm. Tripartite toxin complexes (Tc) belong to the latter class and are widespread in insect and human pathogens[1,2]. Originally discovered in the insect pathogen *Photorhabdus luminescens*[3], many gene loci encoding these proteins have since been found in other organisms. Tc toxins appear to be particularly well represented in enterobacteria, with examples being the insect pathogen *Xenorhabdus nematophila*[4,5], the facultative human pathogen *Photorhabdus asymbiotica*[6], and the deadly human and insect pathogens *Yersinia spp.*[7,8].

Tc toxins consist of three components: TcA, TcB, and TcC. The ~ 1.4 MDa TcA is a homopentameric bell-shaped molecule that mediates target cell association, membrane penetration, and toxin translocation[9]. TcA consists of a central, pre-formed α-helical channel connected to an enclosing outer shell by a linker that acts as an entropic spring during toxin injection[10,11]. The shell is composed of a structurally conserved α-helical domain that is decorated by a neuraminidase-like domain, as well as by variable immunoglobulin-fold receptor-binding domains. In some Tc toxins, the latter are functionally replaced by small soluble proteins that form a quaternary complex with the TcA subunit[12,13].

TcB and TcC together form a ~ 250 kDa cocoon that encapsulates the autoproteolytically cleaved ~ 30 kDa C-terminal hypervariable region (HVR) of TcC, the actual cytotoxic component of the Tc toxin[10,14]. The HVR resides in a partially or completely unfolded state in the cocoon[10,15]. Binding of TcB-TcC to TcA and the subsequent pH-dependent prepore-to-pore transition of the ABC holotoxin result in a continuous translocation channel from the TcB-TcC lumen across the target cell membrane into the cytoplasm that allows the translocation of the HVR[16].

Previously, we resolved several crucial steps of the Tc intoxication mechanism[17]. The first of these is holotoxin formation, where the key feature is a conformational transition of the TcB domain that binds to TcA. This domain is a six-bladed β-propeller, and upon contact of TcA with TcB, the closed blades of the β-propeller unfold and refold in an open form. Consequently, the HVR passes through the β-propeller and enters the translocation channel[16]. The assembled holotoxin binds to receptors on the target cell surface and is endocytosed[10,18]. Upon acidification of the late endosome, the bottom of the TcA shell opens and the prepore-to-pore transition of the Tc toxin occurs[9]. During this process, the compaction of the stretched linker between the shell and the channel drives the channel through the now open bottom of the shell and across the membrane[11]. The α-helical domain of the outer shell, which possesses a stabilizing protein knot, functions as a stator for the transition[19]. Subsequently, the tip of the channel opens and the HVR is translocated into the target cell cytoplasm, where it interferes with critical cellular processes, ultimately causing cell death[18].

Two HVRs from *Photorhabdus luminescens* TcC proteins have been found to function as ADP-ribosyltransferases targeting actin (TccC3HVR) and Rho GTPases such as RhoA and Cdc42 (TccC5HVR)[18]. However, no HVR structures have been solved so far, limiting our understanding of the structural requirements for proteins translocated by Tc toxins. Although previous studies on other bacterial ADP-ribosyltransferases have shown that these enzymes can in fact be structurally similar even without any significant sequence similarity[20–23], we do not know if this also holds true for Tc toxin HVRs.

In this study, we raise an interesting question related to this topic: can the sophisticated Tc toxin translocation system be hijacked and used to transport proteins other than the natural HVRs? Such a proof of concept has already been demonstrated for the anthrax toxin, which was used to transport proteins fused to anthrax lethal factor into cells, including the TccC3HVR of Tc toxins[24]. In fact, similar designs based on the diphtheria toxin and *Pseudomonas aeruginosa* exotoxin A have already been explored as anticancer drugs[25,26]. These systems however have the disadvantage that the fused cargo is exposed to the external environment during delivery, potentially causing the cargo to show premature and unspecific activity and limiting the usefulness of such constructs for both medical and research applications. This drawback could be avoided if the cargo were to be transported to its destination in an inactive form inside the TcB-TcC cocoon, and only activated after protein translocation through the TcA pentamer.

To achieve this, the cargo protein to be translocated would have to be fused to the C-terminus of TcC instead of the native HVR in order to get encapsulated in the cocoon and subsequently translocated through TcA into the target cell (Fig. 1a, b). We explore these aspects here by swapping the TccC3HVR to comparably sized proteins with diverse origins and functions (Supplementary Fig. 1a), and then assess holotoxin formation and cargo translocation of these constructs. We find that no stable ABC holotoxin is formed for cargos below a total size of ~ 20 kDa, which is in accordance with our previous finding that an empty TcB-TcC cocoon does not form ABC with high-affinity either[16]. We then screen different cargos for their translocation after triggering the prepore-to-pore transition in vitro. Several, but not all small proteins are successfully translocated when fused with parts of TccC3HVR. Generally, the non-translocated cargos have a considerably lower isoelectric point (pI) than the translocated constructs. Furthermore, the crystal structures of two non-translocated constructs show us that the formation of structural elements inside the TcB-TcC cocoon, in particular those that interact stably with the inner surface of the cocoon, also prevent cargo translocation.

Together, our results show that cargo proteins must fulfill three prerequisites to be successfully translocated by TcA. The first is the cargo size, which needs to be above a threshold of ~ 20 kDa to form a stable holotoxin complex. The second is the net charge, which must be positive at neutral pH values. The third is that the cargo must not form structural elements within TcB-TcC. Observing these guidelines is the key to creating functional Tc-based protein injection devices.

## Results

**TcC-HVRs are exchangeable in Tc toxins**. As an initial proof that different cargos fused to TcC can be translocated, we tested whether HVRs from different TcC proteins are exchangeable and result in functional, toxic ABC complexes. For this, we replaced the TccC3HVR sequence after the autoproteolytic cleavage site in TcdB2-TccC3 to that of TccC5HVR, resulting in the chimeric TcdB2-TccC3-TccC5HVR complex. After assembly of the ABC-TccC5HVR holotoxin, we assessed cytotoxicity on HEK293T cells. Complete cell death occurs upon addition of 2 nM ABC-TccC5HVR toxin, compared with cell death at 0.5 nM when exposed to the ABC-TccC3HVR holotoxin (Fig. 1c). This is in accordance with previous findings that the cytotoxic effect of TccC5HVR is less pronounced than that of TccC3HVR[18]. This experiment shows that the cocoon formed by TcdB2-TccC3 is capable of also encapsulating and translocating other HVRs, such as TccC5HVR. We therefore chose TcdB2-TccC3 to function as a cocoon scaffold for translocation of other cargo proteins, which are not components of the Tc toxin system. Importantly, the two different HVRs do not show any pronounced sections of sequence

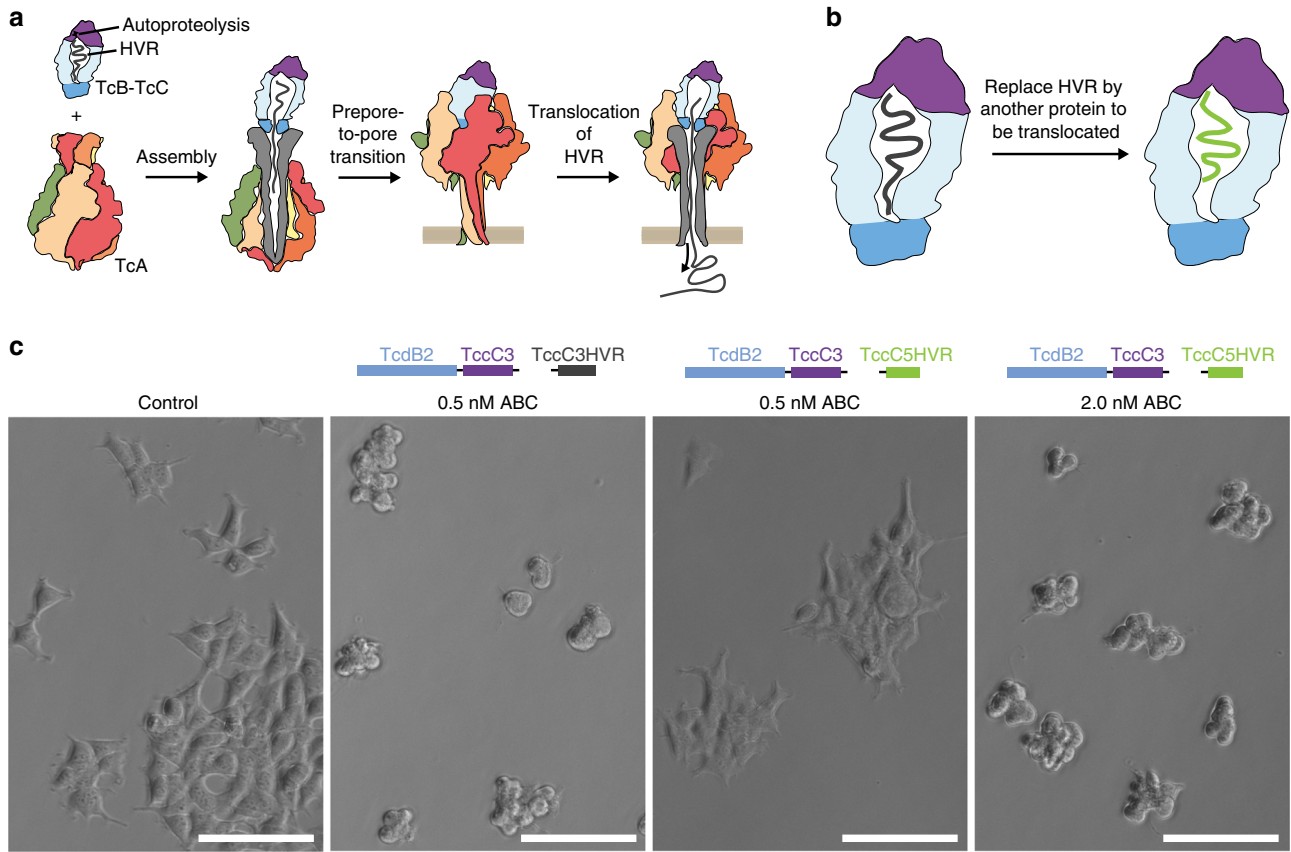

**Fig. 1** Tc toxin translocation mechanism and exchange of TccC HVRs. **a** Schematic of the Tc toxin translocation mechanism. The cocoon-like TcB-TcC component (blue-purple) encapsulates the autoproteolytically cleaved cytotoxic C-terminus of TcC known as the HVR (black). Upon binding of the TcB-TcC component to the pentameric TcA component (visible monomers in red, beige, orange, and green) via the TcA-binding domain of TcB (darker blue), the HVR is released into the central channel of the toxin (gray). Upon pH-induced prepore-to-pore transition at the cell membrane, the channel opens and the HVR is translocated into the cytoplasm. **b** Schematic of the experimental concept. The HVR in the cocoon is recombinantly replaced by an alternative protein to be translocated (green). **c** Effect of TccC3HVR to TccC5HVR replacement in the TcdB2-TccC3 cocoon on cytotoxicity. The ability of the TccC5HVR construct to kill HEK293T cells demonstrates that it is able to effectively translocate through the pore formed by the TcdA1 pentamer. A fourfold higher concentration of the TccC5HVR construct is needed to obtain a cytotoxic effect comparable to that of the original TccC3HVR. While this may indicate less-efficient translocation, the effect is more likely owing to TccC5HVR being a less-potent toxin than TccC3HVR, a finding confirmed by previous studies[18]. Experiments were performed in triplicates with qualitatively identical results. Scale bars: 100 μm

identity (Supplementary Fig. 1b), indicating that there is no conserved motif in Tcc-HVRs that is a general prerequisite for successful translocation.

**Replacement of TcC HVR by heterologous cargo proteins.** Our next step was to replace the HVR of TccC3 with unrelated heterologous proteins and test the capability of the holotoxin to translocate these. The criteria used to select replacement proteins were (i) a small size (11–34 kDa) to guarantee that they fit into the TcB-TcC cocoon, (ii) diverse folds to assess whether this influences the translocation capability, (iii) different oligomeric arrangements to see if this affects proper cocoon assembly, and (iv) various organismal origins to further reduce bias. The proteins selected according to these criteria were the small GTPase cell division control protein 42 homolog (Cdc42) from *Homo sapiens*[27], the herpesviral infected cell protein 47 (ICP47)[28], the light/oxygen/voltage-sensing domain of the *Arabidopsis thaliana* blue light receptor (iLOV)[29], the multi-ligand binding enzyme dihydrofolate reductase from *Escherichia coli* (DHFR)[30], the Ras-binding domain of CRAF kinase from *Homo sapiens* (RBD)[31], and tobacco etch virus (TEV) protease[32]. Several of these proteins possess interesting properties that were hypothesized to provide

additional information on requirements for translocation: Cdc42 is a homodimer in solution[33], ICP47 is intrinsically disordered, iLOV has a flavin mononucleotide chromophore, and TEV contains two β-barrels, which represent stable folds[34] (Supplementary Fig. 1a).

Interestingly, all tested chimeric cocoons could be well expressed in *E. coli* and purified. We then mixed the cocoons with TcA and assessed holotoxin formation by negative stain EM after size exclusion chromatography (SEC). In case of the wild-type TccC3HVR, the affinity of TcB-TcC to TcA is in the picomolar range, resulting in almost complete holotoxin formation[16]. This was also the case for holotoxins composed of TcA and cocoons containing Cdc42 (20.3 kDa) and TEV (28.1 kDa) (Fig. 2a, b, Supplementary Fig. 2a). This demonstrates that TcB-TcC complexes with non-native cargos are able to form holotoxin complexes. In contrast, holotoxin formation was tremendously reduced when cocoons with ICP47 (11.3 kDa), iLOV (13.2 kDa), and DHFR (18.4 kDa) as cargos were used, indicating that there is a size limit for the cargo (Fig. 2b, Supplementary Fig. 2a). Previously, we demonstrated that the HVR inside the cocoon has an influence on the gatekeeper domain of the β-propeller and an empty cocoon has a much lower affinity to TcA than the wild-type[16]. We have proposed that this might be caused by steric

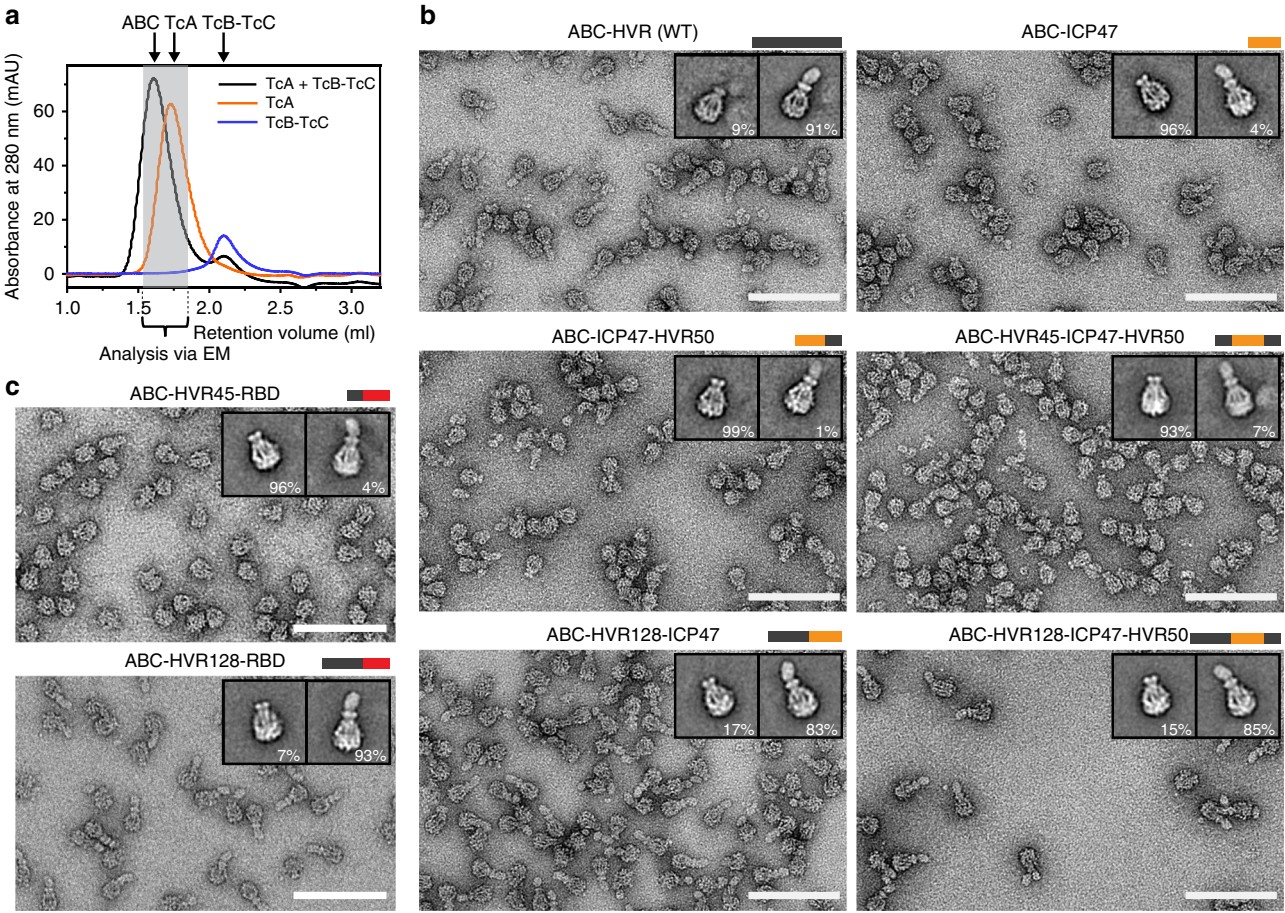

**Fig. 2** ABC holotoxin formation requires a cargo size above a distinct threshold. **a** Comparison of analytical size exclusion chromatography profiles for ABC holotoxin formed by mixing TcA (600 nM) with TcB-TcC (1.2 μM), separate TcA (600 nM), and separate TcB-TcC(WT) (600 nM). The fractions indicated by the gray bar were pooled and analyzed by EM. **b** Negative stain electron micrographs of holotoxins formed by TcA and TcB-TcC with the indicated cargos (WT and ICP47 constructs). Insets: 2D class averages of representative TcA pentamers and holotoxins, with the percentages of particles in the class averages shown below. **c** Negative stain micrographs of holotoxins formed by TcA and TcB-TcC with RBD cargos, with 2D class averages like those described in **b**. While the cargo HVR45-RBD does not trigger holotoxin formation, the larger HVR128-RBD results in assembled holotoxins. Scale bars in **b** and **c**: 100 nm

pressure applied by the HVR and consequently no or a small HVR would result in reduced complex formation. Our results with the different cargos support this hypothesis and indicate that the minimal size requirement for the cargo is around 20 kDa in order to guarantee high-affinity holotoxin assembly.

**The size of the cargo determines holotoxin formation**. To further explore the influence of cargo size on holotoxin formation, we created a truncated version of the native TccC3HVR (residues 1–132) and increased the size of the shorter cargos by adding differently sized parts of TccC3HVR to the N- or C-termini (see Methods for details). To achieve this, we introduced an EcoRI restriction site in frame after the first 138 or 384 base pairs (bp) of the sequence coding for the HVR, resulting in two possible N-terminal extensions of the cargo proteins with 45 or 128 residues from the original HVR, respectively. In addition, we introduced a NotI restriction site 150 bp upstream of the stop codon, resulting in a C-terminal extension of the cargo with the C-terminal 50 residues of the HVR. Altogether, five different combinations of extensions of the cargos were possible: HVR45-cargo, HVR128-cargo, HVR45-cargo-HVR50, HVR128-cargo-HVR50, and cargo-HVR50.

As ICP47 has almost no secondary structure that could influence the size dependency (Supplementary Fig. 1a), it was a

particularly compelling test case. Similarly to the cocoon with only ICP47 (11.3 kDa), cocoons containing the chimeras ICP47-HVR50 (17.1 kDa) or HVR45-ICP47-HVR50 (21.6 kDa) did not have a high affinity to TcA. The same was true for HVR45-RBD (14.2 kDa). However, the longer constructs, namely HVR128-RBD (22.4 kDa), HVR128-ICP47 (24.5 kDa) and HVR128-ICP47-HVR50 (30.3 kDa), resulted in high-affinity holotoxin formation (Fig. 2a–c). Increasing the size of iLOV (HVR128-iLOV (26.4 kDa)) and DHFR (HVR128-DHFR (31.6 kDa)) had the same effect (Supplementary Fig. 2a).

In all chimeras that led to high-affinity holotoxin assembly, the cargo was fused to HVR128. Therefore, one might ask whether these first 128 residues of TccC3HVR contain a motif important for activating TcB-TcC, which the holotoxin-forming Cdc42 and TEV constructs coincidentally possess. However, since HVR (1–128) alone does not activate the cocoon, as can be seen with the HVR (1–132) truncation construct (13.8 kDa) of TccC3HVR (Supplementary Fig. 2b), we believe that the presence of HVR128 is not a prerequisite for the activation mechanism. Taken together, these results confirm our hypothesis that the cargo has to have a certain size (at least ~ 20 kDa) in order to activate the cocoon and put it into an assembly-competent state. At the same time the variety of assembly-competent constructs suggests that the nature of the cargo is not important for this activation

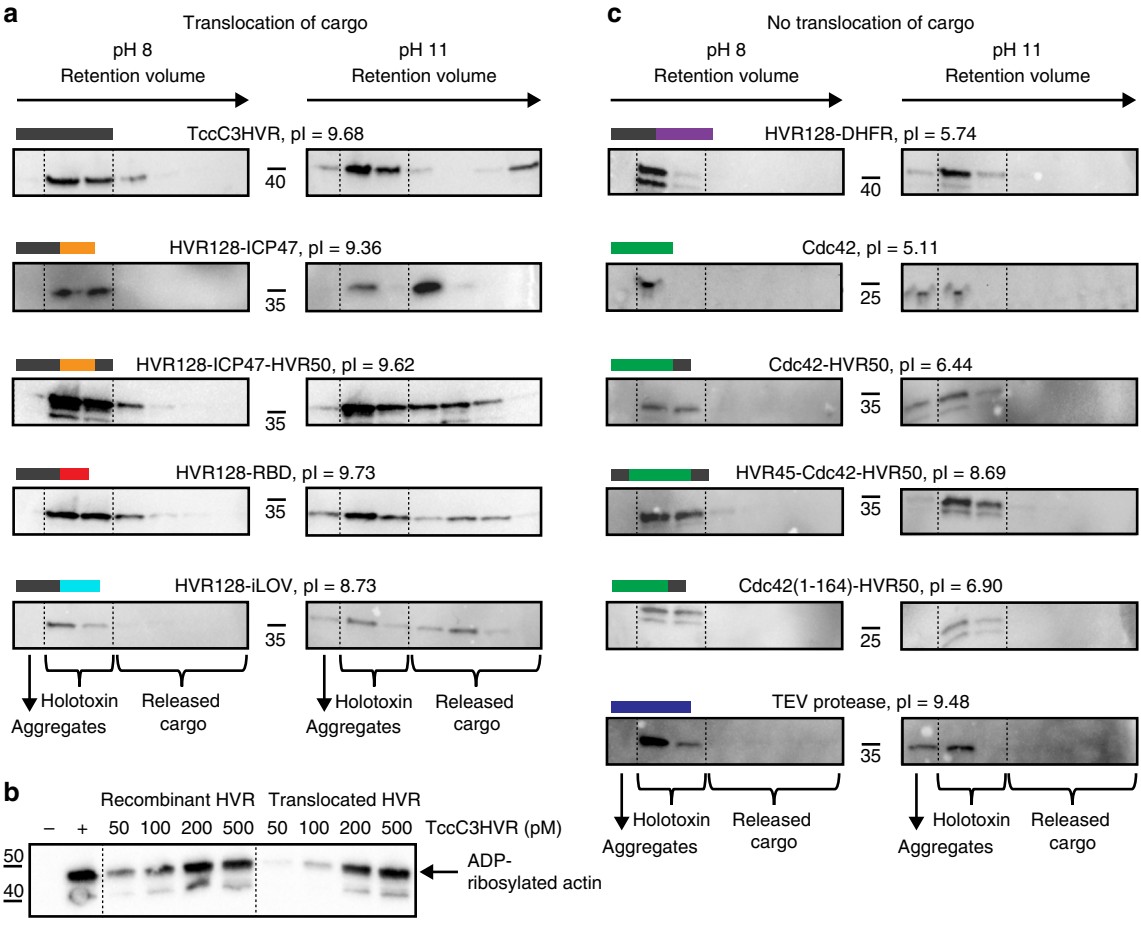

**Fig. 3** Translocation of various non-natural cargos in vitro and activation of TccC3HVR. **a** Successful translocation of ICP47, RBD, and iLOV fused to the N- or C-terminus of TccC3HVR. After incubation at pH 11, which causes the toxin to undergo prepore-to-pore transition, Western blots show that these constructs migrate at higher retention volumes during size exclusion chromatography in comparison with pH 8-incubated controls. This corresponds to translocation and release of the cargo proteins from the holotoxin, as illustrated schematically in Supplementary Fig. 3. **b** ADP-ribosylation of F-actin by translocated TccC3HVR in comparison with recombinantly purified TccC3HVR. Western blots show the increasing appearance of ADP-ribosylated actin dependent on the TccC3HVR concentration for both preparations. −: no TcCC3HVR, +: 100 nM recombinantly purified TccC3HVR. **c** No translocation was observed for Cdc42 and TEV protease cargo alone, or DHFR and Cdc42 fused to the N- or C-terminus of TccC3HVR. Western blots show the presence of the constructs at lower retention volumes during size exclusion chromatography both at pH 8 and pH 11, meaning they co-localize with the rest of the holotoxin even after prepore-to-pore transition and are not translocated. Uncropped images of the Western blots in panels **a**–**c** are provided as a Source Data file

mechanism, supporting the idea that a general high steric 'pressure' is sufficient.

**Translocation of cargo proteins by TcA**. Having demonstrated that holotoxins containing heterologous cargos can be assembled, our next step towards using the Tc scaffold as a customized protein injection system was to show that the cargo can be successfully translocated. To assess the translocation of different cargo proteins without having to rely on protein-specific enzymatic activity read-outs, we developed a cell-free in vitro translocation assay. First, the prepore-to-pore transition of ABC is triggered by shifting the pH to 11. If the cargo can be translocated, it will be ejected through the TcA translocation channel and dissociate from the holotoxin, as described for wild-type ABC (ABC(WT))[15]. Successfully translocated 20–30 kDa cargos can then be easily separated from the 1.7 MDa ABC injection machinery by SEC (Supplementary Fig. 3a), whereas non-translocated cargos will still be holotoxin-associated and therefore co-migrate with the 1.7 MDa peak (Supplementary Fig. 3b). As proof of principle, we first assessed the release of TccC3HVR in ABC(WT) before moving on to test the heterologous

cargos. Indeed, after 48 h of incubation at pH 11, a substantial fraction of TccC3HVR migrates much later than the ABC peak (Fig. 3a), indicating that it has been successfully released and translocated through TcA.

To test whether the TccC3HVR folds and adopts an active conformation after in vitro translocation, we compared the activity of the translocated ADP-ribosyltransferase with that of recombinantly purified TccC3HVR, using an actin-ribosylating assay[35,36] (Methods). Although the activity of the translocated ADP-ribosyltransferase was about half of that of the recombinantly purified enzyme, it clearly ADP-ribosylated F-actin (Fig. 3b). This demonstrates that most of the translocated ADP-ribosyltransferase was folded to its active form after translocation. Importantly, this in vitro assay shows that folding and activity of the ADP-ribosyltransferase do not require additional cofactors such as chaperones and we believe that this is also the case for other translocated cargos.

Next, we tested whether the various cargo proteins could be translocated. We first assessed Cdc42 and TEV, which do not need to be fused to TccC3HVR fragments in order to form a holotoxin. However, the proteins were not translocated by TcA

(Fig. 3c). To determine whether fusing sequences from TccC3HVR can restore translocation competence, we extended Cdc42 with either the C-terminus of TccC3HVR or with both of its termini. However, these cargos could also not be translocated by TcA (Fig. 3c), indicating that neither the C-terminus nor both termini of the HVR are sufficient to enable translocation.

However, the four other cargos that facilitated holotoxin formation only when fused to fragments of TccC3HVR were successfully translocated, namely HVR128-ICP47-HVR50, HVR128-ICP47, HVR128-iLOV, and HVR128-RBD (Fig. 3a). As the C-terminal region that is translocated first in ABC(WT)[16] differs considerably between these cargos, we conclude that there is no specific sequence at the C-terminus that determines whether a cargo is translocated or not. In the case of HVR128-iLOV, iLOV did not produce fluorescence in the cocoon, whereas TcB-TcC-iLOV did (Supplementary Fig. 3d), indicating that HVR128-iLOV is stored in an unfolded form. We conclude that extending the N-terminus by 128 additional residues prevents the folding of HVR128-iLOV inside the cocoon, whereas the fluorescence of TcB-TcC-iLOV shows that iLOV without the extension is able to fold inside the cocoon and has access to the cofactor FMN.

In contrast to the other three HVR128-containing cargos, DHFR is not translocated when fused to the same HVR128 N-terminus (Fig. 3c). Together with the non-translocated HVR45-Cdc42-HVR50 cargo, this shows that the sequence of the N-terminus is also not the determinant of cargo transport through TcA. Therefore, there must be another factor at work that establishes translocation competence.

A comparison of the four translocated fusion proteins (HVR128-ICP47, HVR128-ICP47-HVR50, HVR128-RBD, and HVR128-iLOV) and the native cargos Tcc3HVR and Tcc5HVR shows that their common feature is a positive net charge at neutral pH, with isoelectric points of at least 7.9 (Fig. 3a). In the case of the fusion constructs, this is mainly due to the highly positively charged HVR128 (pI 9.75), which is larger than the cargo protein in all cases (maximum size 13.2 kDa). We therefore conclude that besides being large enough, the cargo has to be positively charged (pI ≥ ∼ 8) in order to be translocated.

In line with this, the six constructs that formed holotoxin complexes but did not show translocation had mostly negatively charged cargos (Fig. 4). Only two non-translocated cargos, namely HVR45-Cdc42-HVR50 and TEV were positively charged. The TEV construct we used has a highly positively charged penta-arginine tail to allow purification by cation exchange chromatography without changing its activity[37], a modification that raises the pI of TEV from 8.67 to 9.62. Although such a change in pI should favor translocation, it is possible that distributing the positive charges in such an uneven manner rather hinders it.

TEV protease is not an exceptionally stable protein, as it unfolds at ∼ 52 °C[38]. However, it contains two small β-barrels (Supplementary Fig. 1a), which are known to be a notoriously stable fold[34]. Therefore, we believe that these domains are more rigid than the native Tcc3HVR or other translocated cargos such as HVR128-iLOV, which might prevent the protein from being translocated. To prove this, we determined the structure of TcB-TcC-TEV to a resolution of 3.7 Å using X-ray crystallography (Supplementary Table 1). The cocoon of TcB-TcC-TEV shows an identical size and shape to the TcB-TcC(WT) cocoon[10], with a $C_\alpha$ RMSD of 0.552 Å². However, the electron density inside appears already at a higher sigma level, indicating that the encapsulated TEV is indeed more rigid and ordered than the native TccC3HVR (Supplementary Fig. 3c). This is the likely cause for TEV not being translocated. As folded proteins do not fit through the narrow constriction site of the translocation channel[16], the formation of defined structural elements such as β-barrels in the cocoon probably results in translocation arrest.

| | Cargo | Size (kDa) | High-affinity ABC formation | pI | Translocation |
|---|---|---|---|---|---|
| | TccC3HVR | 30.4 | yes | 9.68 | Yes |
| | TccC5HVR | 30.3 | yes | 7.90* | Yes, in vivo** |
| | ICP47 | 11.3 | no | 5.64 | n.a. |
| | ICP47-HVR50 | 17.1 | no | 8.09 | n.a. |
| | HVR45-ICP47-HVR50 | 21.6 | no | 9.73 | n.a. |
| | HVR128-ICP47 | 24.5 | yes | 9.36 | Yes |
| | HVR128-ICP47-HVR50 | 30.3 | yes | 9.62 | Yes |
| | HVR45-RBD | 14.2 | no | 10.06 | n.a. |
| | HVR128-RBD | 22.4 | yes | 9.73 | Yes |
| | iLOV | 13.2 | no | 5.13 | n.a. |
| | HVR128-iLOV | 26.4 | yes | 8.73 | Yes |
| | DHFR | 18.4 | no | 4.77 | n.a. |
| | HVR128-DHFR | 31.6 | yes | 5.74 | no |
| | Cdc42 | 20.3 | yes | 5.11 | no |
| | Cdc42-HVR50 | 26.1 | yes | 6.44 | no |
| | HVR45-Cdc42-HVR50 | 30.6 | yes | 8.69 | no |
| | Cdc42(1-164)-HVR50 | 24.9 | yes | 6.90 | no |
| | TEV | 28.1 | yes | 9.48 | no |

**Fig. 4** List of cargos tested in this study. The nomenclature of the chimeras, for example, HVR128-ICP47-HVR50, indicates how many N- or C-terminal TccC3HVR residues have been pre- or appended to the cargo protein. The colored bars indicate the composition of the constructs and are used consistently in all figures. n.a.: not applicable. *The pI of TccC5HVR is changed from 8.65 to 7.90 by the addition of four residues (MPEF) to the N-terminus, resulting from cloning. **Toxicity to HEK293T cells (see Fig. 1c)

**Interaction with cocoon inhibits translocation of Cdc42.** Although Cdc42 does not possess any folds that are immediately classifiable as very stable, we examined the Cdc42 constructs in more detail and solved the crystal structure of TcB-TcC-Cdc42 to 2.0 Å to find out whether Cdc42 folding inside the cocoon is nonetheless a plausible explanation for its translocation incompetence (Fig. 5a, Supplementary Table 1).

The overall shape of the TcB-TcC-Cdc42 cocoon is identical to wild-type (TcB-TcC(WT)) and empty TcB-TcC[10,16], with $C_\alpha$-RMSD values of 0.424 A² and 0.414 Å² between TcB-TcC-Cdc42/TcB-TcC(WT) and TcB-TcC-Cdc42/empty TcB-TcC, respectively, indicating that a different cargo does not influence the RHS repeat structure of the cocoon. Similarly to TccC3HVR in the TcB-TcC (WT) cocoon, Cdc42 is not structured inside the cocoon. However, we found an ordered electron density in close spatial proximity to the β-propeller domain, which corresponds to an α-helix not present in other TcB-TcC structures obtained so far. The high-resolution of the map and availability of the Cdc42 structure[27] enabled us to identify the α-helix inside the cocoon as the C-terminus of Cdc42 (N167—P179) (Supplementary Fig. 4a, b). Importantly, the helix is attached to the TcB protein in an orientation perpendicular to that expected from natural translocation (Fig. 5a, b), and its amphipathic nature facilitates interaction with a hydrophobic pocket on the inner surface of the cocoon (Fig. 5c). The side chains of F169, I173, L177, and P179 are rigidly oriented toward the hydrophobic cleft, where they are in close contact with L39, L41, P42, L366, L368, M702, N704, V708, H710, L1203, and F1349 of the cocoon (Fig. 5d). In contrast, D170, E171, and E178 face the TcB-TcC lumen with more degrees of freedom. This is reflected in the quality of the crystallographic density, with only the residues pointing toward the cocoon surface being well resolved (Supplementary Fig. 4c).

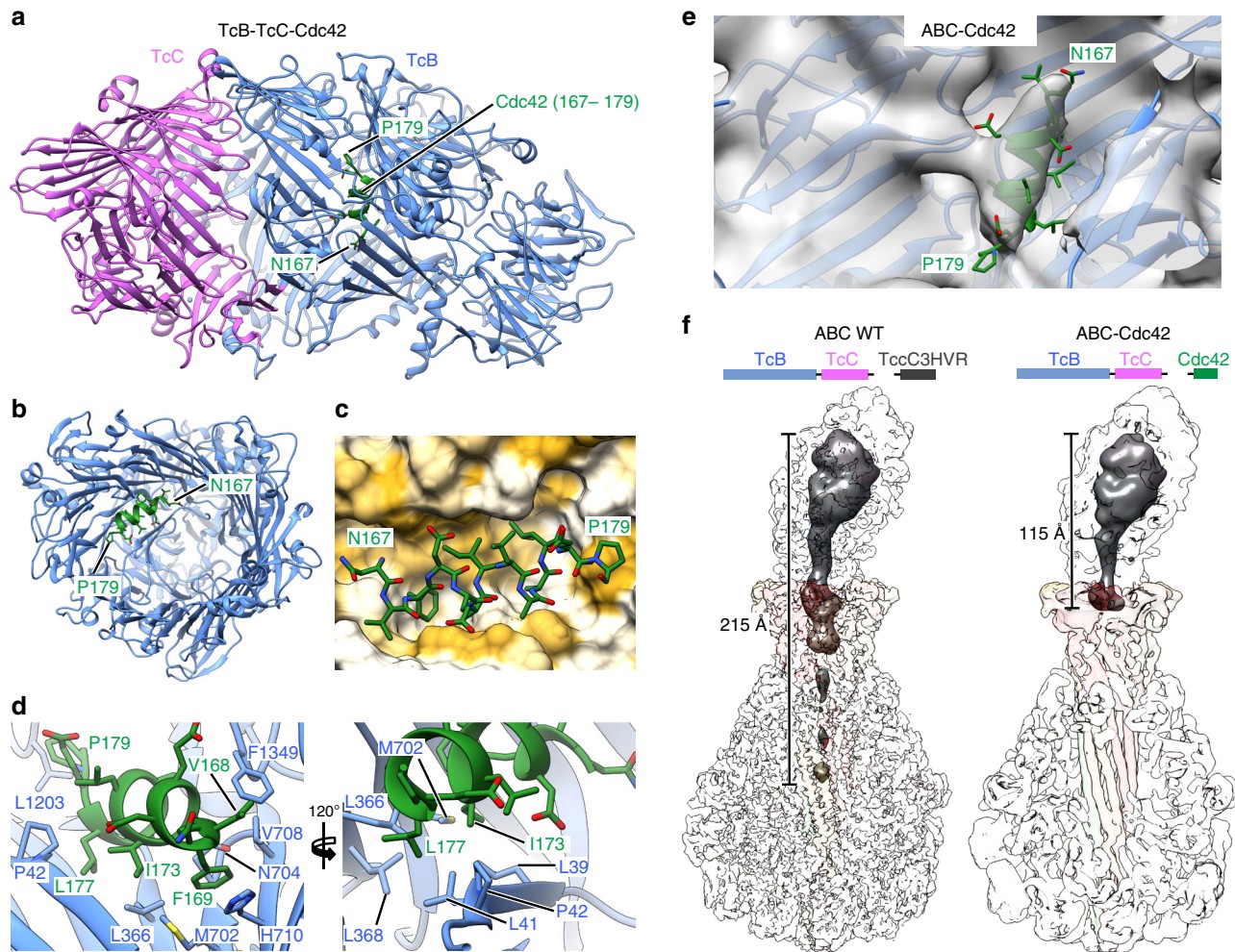

**Fig. 5** Structure of TcB-TcC-Cdc42 alone and in complex with TcA. **a** Crystal structure of the TcB-TcC-Cdc42 complex. The C-terminal region of Cdc42 (N167–P179, green) forms a helix inside the cocoon. **b** Top view into TcB-TcC-Cdc42, with the TcC model removed for illustrative purposes. The C-terminal Cdc42 α-helix (green) can be seen attached to one side of the cocoon, whereas remaining Cdc42 density is too disordered to resolve. **c** Surface representation of the binding pocket of the hydrophobic C-terminal Cdc42 helix in the TcB section of the cocoon. The molecular surface of TcB is colored according to its hydrophobicity[52], with hydrophobic regions shown in ochre and polar regions in white. **d** Model of the C-terminal Cdc42 α-helix in the binding pocket of TcB. Two orientations are shown. Side chains that face the Cdc42 α-helix are indicated. **e** Rigid-body fit of the crystal structure of TcB-TcC-Cdc42 into the cryo-EM density map of ABC-Cdc42. Density corresponding to the α-helix of Cdc42 is also present in the cryo-EM structure of the holotoxin at the same site. **f** Comparison of cryo-EM density reconstructions of ABC(WT) (PDB 6H6F) and ABC-Cdc42 (this work). The surfaces of the TcA, TcB, and TcC subunits are transparent, and the density corresponding to the ADP-ribosyltransferase TccC3HVR (WT, left) and Cdc42 (right) is dark gray. The latter is shown at a lower threshold and filtered to 15 Å resolution

Interestingly, the affinity of the Cdc42 α-helix for this part of the cocoon is strong enough to displace the N-terminus of TcB, which resides at this position in TcB-TcC(WT) (Supplementary Fig. 4d). The N-terminus of TcB-TcC-Cdc42 is correspondingly not resolved, indicating that it protrudes into the cocoon lumen. The tight attachment of Cdc42 via its C-terminus in close spatial proximity to the TcB gatekeeper domain[16] could help this construct to form a holotoxin with high affinity comparable to the constructs with larger cargos, despite Cdc42 being at the lower limit of the cargo size prerequisite (Fig. 2, Supplementary Fig. 2).

The stable nature of the Cdc42 α-helix interaction with the cocoon raises the question of whether it remains bound even after holotoxin formation, in which case it would not be able to enter the translocation channel. We addressed this issue by determining the 5 Å structure of the ABC-Cdc42 holotoxin using cryo-EM (Supplementary Fig. 5). Indeed, a small helix-shaped density appears at the interaction site even at a comparably high-map binarization threshold, despite the limited resolution of the 3D

reconstruction. Fitting the entire crystal structure of TcB-TcC-Cdc42 into the cryo-EM map results in a very good match of the Cdc42 α-helix with the additional map density of ABC-Cdc42 (Fig. 5e). In contrast, no comparable cryo-EM density is present at the same position in ABC(WT) and an aspartyl protease deficient variant[16] (Supplementary Fig. 6), indicating that the cargo does not form any structural elements at this location in functional holotoxins.

The tight association of the C-terminal α-helix to the cocoon results in Cdc42 translocation arrest already at an early stage. If the C-terminus that would normally be translocated first through TcA[16] is bound elsewhere, then why does Cdc42 form a holotoxin? Analysis of the cryo-EM data shows that no additional density is present in the TcA translocation channel, unlike in ABC(WT) (Fig. 5f). There is however a stretch of Cdc42 density that protrudes into the β-propeller domain of TcA but does not continue further into the channel. This indicates that although Cdc42 cannot be translocated, it applies a steric pressure on the

TcB gatekeeper domain, resulting in the high-affinity binding of TcB to TcA.

These results demonstrate that if the encapsulated cargo forms stable structural elements that strongly interact with the inner lumen of the cocoon, the cargo cannot be translocated by TcA even if it is positively charged and large enough.

## Discussion

Taken together, our results indicate that the *P. luminescens* Tc toxin can be successfully transformed into a universal protein translocation system as long as the cargo protein fulfills several prerequisites. The first of these is cargo size. While its upper limit is defined by the size of the cocoon, an exact value was not determined in this study. The largest cargo tested here was HVR128-DHFR (31.6 kDa), which is 1.2 kDa larger than the TccC3HVR natural cargo (Fig. 4). At the same time, the largest natural cargo of a *P. luminescens* Tc toxin is TccC1HVR at 35.1 kDa, providing a potential upper size limit and meaning that larger ADP-ribosyltransferases like the 49.4 kDa C2 toxin of *Clostridium botulinum*[21] would likely not fit into the cocoon. It remains to be explored whether the cocoon itself can be enlarged by adding further RHS repeats, thereby expanding the upper cargo size limit.

Importantly, we discovered that the lower size limit of the cargo is in the 20–22 kDa range, and the ability to form the ABC holotoxin is drastically reduced if the cargo size is below this limit (Fig. 4, Fig. 6). This finding echoes previous results showing that a comparable decrease in affinity between empty TcB-TcC and TcA occurs because there is no HVR present to apply steric pressure on the TcB gatekeeper domain that is essential for binding to TcA[16]. Similarly, small cargos will likely be too mobile inside the cocoon to cause gatekeeper destabilization. The lower size limit is slightly flexible, e.g. Cdc42 (20.3 kDa) forms holotoxin whereas HVR45-ICP47-HVR50 (21.6 kDa) does not (Fig. 4). This is potentially caused by the C-terminal helix of Cdc42 attaching the rest of the cargo to the vicinity of the cocoon exit, allowing the Cdc42 N-terminus to destabilize the gatekeeper more easily, which results in holotoxin formation but also prevents subsequent cargo transport. We therefore hypothesize that translocation will likely be impaired if a TcB-TcC cocoon loaded with a small cargo binds with high affinity to TcA.

The second prerequisite for successful transport is a net positive charge of the cargo protein. All naturally occurring HVRs of *P. luminescens* Tc toxins have a pI of at least 7.9 (TccC5HVR), and the protein with the lowest pI for which we observed translocation in our in vitro assay is HVR128-iLOV with a pI = 8.73 (Fig. 4). Since the gatekeeper constriction site that serves as the entry point into the TcA channel is formed by several acidic residues[16], negatively charged constructs might be repelled. In addition, the translocation channel of TcA has several negatively charged bands[10,11] that facilitate the transport of cations but not anions[10,39]. It is unlikely that the translocation behavior is different when the cargo is translocated across a native membrane in comparison with the in vitro system, as TcA forms a translocation channel through the membrane that is laterally closed and does not allow the entry of lipids into the channel lumen[11]. The transport of the native TccC3HVR across lipid nanodiscs occurs without the need for a proton gradient or a chaperone[15], although a facilitation of translocation by the latter two factors cannot be excluded in the context of a cellular environment. Furthermore, single channel conductivity experiments[9] and cell toxicity assays with TcA in the absence of a TcB-TcC cocoon[40] show that an open pore is formed in the membrane. Consequently, we believe cargo proteins can move unhindered through the transmembrane channel provided their pI is basic enough to do so.

A sufficiently large size and a positive net charge are however not the only points that need to be considered, because some of the larger cargos with high pI were not translocated (TEV and HVR45-Cdc42-HVR50). Therefore, a third prerequisite needs to be fulfilled: the encapsulated cargo must not form any tertiary structures or stably interact with the cocoon. The two above-mentioned cargos violate this prerequisite, with TEV containing two highly stable β-barrels, and Cdc42 possessing an amphipathic C-terminus that associates with a hydrophobic binding pocket in the cocoon.

To summarize, we recommend adhering to the following guidelines in order to successfully turn a Tc toxin into a protein translocation device (Fig. 6): (i) Choose a protein between 20 and 35 kDa to insert after the autoproteolytic cleavage site of TcC. In case the cargo is too small, add N- or C-terminal extensions that do not interfere with protein function or link several copies of the protein together. (ii) Choose a protein with a pI of at least 8.0. Increase the pI by addition of positively charged residues or site-directed mutagenesis if necessary and possible. (iii) Avoid cargo proteins that might form highly stable structures already in the cocoon lumen. (iv) Avoid cargo proteins that contain amphipathic helices or extensive hydrophobic patches that might stably interact with the inner surface of the cocoon.

## Methods

**Source plasmids for cargo proteins.** pGEX-iLOV was a gift from John Christie (Addgene plasmid #26587)[29]. pcDXc3A-AN44-ICP47 containing the coding sequence (CDS) for ICP47 with a C-terminal HA epitope was a gift from Robert Tampé, Goethe University of Frankfurt, Germany. pGEX-RBD with the CDS for the Ras-binding domain (RBD) of CRAF kinase was a gift from Andreas Ernst, Goethe University of Frankfurt, Germany. A customized plasmid with the CDS for dihydrofolate reductase (DHFR) was a gift from Yaowen Wu, Max Planck Institute of Molecular Physiology, Dortmund, Germany. pOPIN-MBP-TEV with the CDS for TEV protease with a C-terminal penta-arginine tail was a gift from the Dortmund Protein Facility (DPF), Max Planck Institute of Molecular Physiology, Dortmund, Germany.

**Cloning of TcB-TcC with non-natural cargo proteins.** We used the fusion protein TcdB2-TccC3[10] as a carrier for heterologous cargo proteins to replace TccC3HVR. We therefore introduced an EcoRI restriction site by site-directed mutagenesis after the codon coding for the conserved P680 of TccC3, which is two residues after the aspartyl protease site[15]. Subsequently, we cloned the cargo proteins in frame via EcoRI and XhoI, resulting in an N-terminal extension of four residues (MPEF). In the case of TccC5HVR, the N-terminal extension results in a change of the pI from 8.65 to 7.90. In the case of Cdc42, the EcoRI site was inserted after the codon for

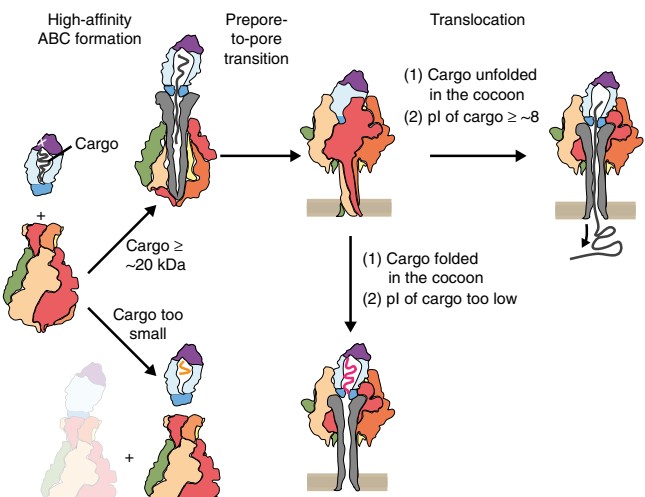

**Fig. 6** Prerequisites for creating a functional customized Tc translocation system. TcA and TcB-TcC assemble into a holotoxin with high affinity only when the cargo is sufficiently large (≥ 20–22 kDa). Translocation upon prepore-to-pore transition only occurs if the cargo is unfolded in the cocoon and its pI is ≥ 8

L678, and the resulting point mutation P680F was reverted by site-directed mutagenesis. Cargo proteins with N-terminal extensions were generated by inserting the EcoRI site 135 or 384 base pairs after the aspartyl protease site and restriction cloning as described above, resulting in 45 or 128 N-terminal residues of TccC3HVR and two residues (EF) encoded by the EcoRI site being attached to the cargo. Cargo proteins with C-terminal extensions were generated by inserting a NotI restriction site 150 base pairs upstream of the stop codon, followed by restriction cloning via EcoRI and NotI. This resulted in a C-terminal extension of the cargo proteins containing 50 C-terminal residues of TccC3HVR and three residues (GGR) encoded by the NotI site. An overview of all resulting TcB-TcC cargos is listed in Fig. 4. An overview of the primers used in this study is shown in Supplementary Table 3.

**Protein production.** *P. luminescens* TcdA1 (TcA) was expressed and purified as described previously[16]. *E. coli* BL21-CodonPlus(DE3)-RIPL cells were transformed with pET19b containing the *tcdA1* gene with an N-terminal hexahistidine tag and a pre-culture was inoculated from a freshly transformed colony. 10 L LB medium were inoculated to an $OD_{600}$ of 0.05 and incubated at 37 °C. At an $OD_{600}$ of 0.6, expression was induced by 25 μM IPTG and carried out overnight at 20 °C. Cells were disrupted in lysis buffer (20 mM Tris-HCl pH 8.0, 200 mM NaCl, 5 mM imidazole, 0.05% Tween-20) using a microfluidizer. After removal of cell debris by centrifugation, the supernatant was applied to a 5 mL Ni-NTA column (GE Healthcare Life Sciences) and washed with 10 column volumes (CVs) of washing buffer (lysis buffer with 50 mM imidazole). Subsequently, the protein was eluted with elution buffer (lysis buffer with 150 mM imidazole). Eluted TcdA1 was dialyzed against 20 mM HEPES-NaOH pH 8.0, 150 mM NaCl, 0.05% Tween-20 and SEC on a Sephacryl S400 16/60 column (GE Healthcare Life Sciences) equilibrated in the same buffer was performed as a final purification step.

All TcB-TcC cargo variants were expressed and purified analogously to WT TcB-TcC as described previously[11]. *E. coli* BL21-CodonPlus(DE3)-RIPL cells were transformed with pET28a encoding the *tcdB2-tccC3* genes (WT or cargo variants) with an N-terminal hexahistidine tag. In total, 5 or 10 L of LB medium containing 30 μM IPTG were directly inoculated with a freshly transformed colony. Cells were grown at 28 °C for 4 h, followed by 25 °C for 20 h and 20 °C for 24 h. Subsequently, cells were disrupted in lysis buffer (20 mM Tris-HCl pH 8.0, 300 mM NaCl, 10% glycerol) using a microfluidizer. After removal of cell debris, the supernatant was applied to a 5 mL Ni-NTA column (GE Healthcare Life Sciences) and washed with 10 CVs of washing buffer (lysis buffer with 40 mM imidazole), followed by elution with a linear gradient from 40 mM to 250 mM imidazole over 10 CVs. The eluted protein was diluted with dilution buffer (20 mM Tris-HCl pH 8.0, 5% glycerol) to a final NaCl concentration of 20 mM and immediately loaded on a 5 mL HiTrapQ column (GE Healthcare Life Sciences) as a second purification step. After washing with 10 CVs of washing buffer 2 (20 mM Tris-HCl pH 8.0, 20 mM NaCl, 5% glycerol), the protein was eluted with a linear gradient from 20 to 500 mM NaCl over 20 CVs. Fractions containing TcB-TcC were subjected to SEC using a Superdex 200 10/300 or a Superdex 200 16/60 column (GE Healthcare Life Sciences) equilibrated in gel filtration buffer (20 mM Tris-HCl pH 8.0, 150 mM NaCl, 5% glycerol) as a final purification step.

The gene coding for TccC3HVR (TccC3(679–960)) was cloned in pET19b in frame with a C-terminal 3C protease cleavage site and a hexahistidine tag. Expression was performed in *E. coli* BL21-CodonPlus(DE3)-RIPL cells. 2 L LB medium were inoculated to an $OD_{600}$ of 0.05 from an overnight pre-culture and incubated at 37 °C. At an $OD_{600}$ of 0.6, expression was induced by 100 μM IPTG and carried out overnight at 20 °C. Cells were disrupted in lysis buffer (20 mM Tris-HCl pH 8.0, 300 mM NaCl, 10 mM imidazole) using a microfluidizer. After removal of cell debris by centrifugation, the supernatant was applied to a 5 mL Ni-NTA column (GE Healthcare Life Sciences) and washed with 10 CVs of lysis buffer, followed by elution with a gradient from 10 to 250 mM imidazole over 15 CVs. Eluted TccC3HVR was dialyzed against 20 mM Tris-HCl pH 8.0, 150 mM NaCl for 2 h before addition of recombinantly purified, hexahistidine-tagged Human Rhinovirus 3C protease (0.05 mg per mg of TccC3HVR), and dialysis was continued for 16 h at 4 °C. Subsequently, the protein solution was again applied to a 5 mL Ni-NTA column and the flow-through containing TccC3HVR without the histidine tag was subjected to SEC on a Superdex 200 16/60 column (GE Healthcare Life Sciences) using gel filtration buffer (20 mM Tris-HCl pH 8.0, 150 mM NaCl).

**ABC holotoxin formation.** Purified TcA (600 nM pentamer) and different TcB-TcC variants (1.2 μM) were mixed together and incubated for at least 1 h at 4 °C before removing the excess of free TcB-TcC by SEC on a Superose 6 Increase 5/150 column (GE Healthcare Life Sciences). For TcB-TcC variants containing HVR-RBD chimeras, 300 nM of TcA pentamer and 600 nM of TcB-TcC were used instead. The formation of the resulting ABC variants was verified by negative stain electron microscopy.

**Negative stain electron microscopy.** After SEC, 3 μL of each ABC variant at 0.1 mg/mL were incubated for 1 min on a glow-discharged 400-mesh copper grid (Agar Scientific) with an additional layer of thin carbon film. Subsequently, the

sample was blotted with Whatman no. 4 filter paper and stained with 0.75% uranyl formate. Images were recorded on a FEI Tecnai Spirit TEM operating at 120 kV and equipped with a F416 CMOS detector (TVIPS). To quantify the ratio of holotoxin assembly, we picked at least 1000 particles from every ABC variant using crYOLO[41] and subjected them to 2D classification with ISAC[42] in SPHIRE[43].

**Cell intoxication.** HEK293T cells were intoxicated with WT ABC or ABC-TccC5HVR. Cells ($5 \times 10^4$ per well) were grown adherently overnight in 400 μL DMEM/F12 medium (Pan Biotech) and 0.5 or 2 nM holotoxin was subsequently added. Incubation was performed for 16 h at 37 °C before imaging. Experiments were performed in triplicate. Cells were not tested for *Mycoplasma* contamination.

**In vitro protein translocation assay.** After ABC formation and removal of unbound TcB-TcC via SEC, 200 nM ABC (WT or cargo variants) was mixed with DDM (final concentration: 0.1%) and dialyzed against 20 mM CAPS-NaOH pH 11.2, 150 mM NaCl, 0.1% DDM for 48 h at 4 °C. As a parallel control experiment, the same amount of ABC with 0.1% DDM was dialyzed against 20 mM Tris-HCl pH 8.0, 150 mM NaCl, 0.1% DDM under the same conditions. Subsequently, the dialyzed proteins were subjected to SEC on a Superose 6 Increase 5/150 column equilibrated in the respective dialysis buffer. SEC fractions corresponding to the exclusion volume (aggregated holotoxin after dialysis), the major peak of the holotoxin, the tail of the holotoxin peak, and the peak after holotoxin were analyzed via sodium dodecyl sulfate-polyacrylamide gel electrophoresis (SDS-PAGE) and Western blot for the presence of the cargo. In the cases where the cargo is translocated and released from the holotoxin, a Western blot signal appears in the fractions after the holotoxin peak at pH 11 (Supplementary Fig. 3a). In the cases where the cargo is not translocated, Western blot signals are only found in fractions containing the holotoxin, both at pH 8 and pH 11 (Supplementary Fig. 3b).

**Western blot and immunodetection.** After SDS-PAGE of the collected SEC fractions (10 μL per fraction) on a 4–15% acrylamide gradient gel, the proteins were transferred onto a PVDF membrane using a Trans-Blot Turbo semi-dry transfer system (Bio-Rad). In the cases where the cargo proteins were fusion constructs with N- or C-terminal parts of TccC3HVR, a custom-made anti-TccC3HVR rabbit polyclonal antibody (Cambridge Research Biochemicals) was used as the primary antibody at 1:1000 dilution. For Cdc42 without a TccC3HVR fusion, an anti-Cdc42 rabbit polyclonal antibody (Cell Signaling Technology, Cat. No. 2462) was used as the primary antibody at 1:1000 dilution. For TEV, an anti-TEV protease rabbit polyclonal antibody (Novus Biologicals, Cat. No. NBP1–97669) was used as the primary antibody at 1:500 dilution. An HRP-conjugated goat anti-rabbit antibody (Bio-Rad, Cat. No. 170–6515) was applied as the secondary antibody at 1:2000 dilution in all cases. Detection was performed with Western Lightning Plus ECL reagent (PerkinElmer, Cat. No. NEL104001EA) and imaged in a ChemiDoc MP imaging system (Bio-Rad).

**Isolation of translocated TccC3HVR.** To isolate translocated TccC3HVR, we assembled ABC wild-type holotoxin (700 nM) as described above, and triggered the prepore-to-pore transition by dialysis against 20 mM CAPS-NaOH pH 11.2, 150 mM NaCl in the presence of 9 μM Msp1D1 nanodisc scaffold protein (Cube Biotech, Cat. No. 26112) and 490 μM 1-palmitoyl-2-oleoyl-sn-glycero-3-phos-phocholine (Avanti Polar Lipids, Cat. No. 850457 C) dissolved in 1% sodium cholate. After 72 h of dialysis at 4 °C, we subjected the holotoxin to SEC on a Superose 6 increase 5/150 column equilibrated in dialysis buffer and collected the fractions after the holotoxin peak (see Supplementary Fig. 3a).

**ADP-ribosylation assay.** For in vitro ADP-ribosylation, rabbit muscle actin was prepared as described previously[44] and filamentous actin (F-actin) was formed by overnight incubation in ADP-ribosylation buffer (50 mM HEPES-NaOH pH 7.3 100 mM KCl, 2 mM $MgCl_2$) at 4 °C. In total, 2 μM of F-actin were mixed with 1 mM $NAD^+$ and different concentrations (50–500 pM) of recombinantly purified or translocated TccC3HVR and incubated for 12 min at 22 °C. The reaction was stopped by addition of SDS-PAGE sample buffer and immediate heating to 95 °C. ADP-ribosylation of F-actin was assessed via SDS-PAGE, Western blot and immunodetection using anti-pan-ADP-ribose binding reagent (Millipore, Cat. No. MABE1016) at 1:2000 dilution and an HRP-conjugated goat anti-rabbit antibody as described above.

**Florescence spectroscopy.** Florescence emission spectra of TcB-TcC(WT), TcB-TcC-iLOV and TcB-TcC-HVR130-iLOV (500 nm protein concentration) were recorded in 20 mM Tris-HCl pH 8.0, 150 mM NaCl, 0.05% Tween-20 using a FluoroMax-4 fluorescence spectrophotometer (Horiba). The excitation wavelength was 450 nm, and emission was recorded from 490 to 620 nm. Purified iLOV with a GST-tag[29] was used as positive control.

**X-ray crystallography of TcB-TcC-Cdc42 and TcB-TcC-Cdc42.** TcB-TcC-Cdc42 and TcB-TcC-TEV were crystallized using the sitting-drop vapor diffusion method

at 20 °C. For TcB-TcC-Cdc42, initially, 2D crystals formed after mixing 1 μL of 10 mg/mL protein solution with 1 μL reservoir solution containing 0.1 M sodium chloride, 0.1 M magnesium chloride, 0.1 M tri-sodium citrate pH 5.5 and 12% PEG 4000. The 2D crystals were used to prepare a seed solution. Final 3D crystals were obtained by mixing 1 μL of 10 mg/mL protein solution with 0.5 μL seed solution and 1.5 μL reservoir solution containing 0.1 m magnesium chloride, 0.1 M tri-sodium acetate pH 4.6 and 12 % PEG 6000. TcB-TcC-TEV crystallized in the same crystallization buffer without seeding. Prior to flash freezing in liquid nitrogen, the crystals were soaked in reservoir solution containing 20% glycerol as a cryoprotectant.

X-ray diffraction data were collected at the PXII-X10SA beamline at the Swiss Light Source (Villigen, Switzerland) using a wavelength of 0.97958 Å and 0.9998 Å for TcB-TcC-Cdc42 and TcB-TcC-TEV, respectively. The X-ray data set was integrated and scaled using XDS[45]. Phases were determined by molecular replacement with PHASER implemented in PHENIX[46] using the crystal structure of WT TcdB2-TccC3 (PDB 4O9X) as a search model. For the TcB-TcC-TEV crystal two data sets were collected with 60° translation and merged together using XSCALE[45]. For phasing, the merged data set was extended to a resolution of 3.3 Å and the refinement was performed with a resolution cutoff of 3.7 Å in PHENIX[46]. TcB-TcC-Cdc42 crystallized in the orthorhombic space group $P2_12_12_1$ with unit cell dimensions of $96 \times 156 \times 179$ Å and one molecule per asymmetric unit. TcB-TcC-TEV crystallized in primitive hexagonal space group $P3_221$ with unit cell dimensions of $234 \times 234 \times 143$ Å and one molecule per AU. The structures were optimized by iteration of manual and automatic refinement using COOT[47] and PHENIX[48] to a final $R_{free}$ of 25% for TcB-TcC-Cdc42 and 34% for TcB-TcC-TEV. Data collection and refinement statistics are summarized in Supplementary Table 1.

**Cryo-EM sample preparation and data acquisition**. In total, 3 μL of 2.1 mg/mL ABC-Cdc42 in 20 mM Tris-HCl pH 8.0, 150 mM NaCl, 0.05% Tween-20 were applied to a glow-discharged holey carbon grid (Quantifoil, QF 2/1, 300 mesh). Subsequently, the sample was vitrified in liquid ethane with a Cryoplunge3 plunger (Cp3, Gatan) using 1.6 s blotting time at 90% humidity and 22 °C.

A data set of ABC-Cdc42 was collected at the Max Planck Institute of Molecular Physiology, Dortmund using a Cs corrected Titan Krios equipped with an XFEG and a Falcon II direct electron detector. Images were recorded using the automated acquisition program EPU (FEI) at a magnification of 59,000× corresponding to a pixel size of 1.14 Å/pixel on the specimen level. In total, 3024 movie-mode images were acquired in a defocus range of 1.0–3.2 μm. Each movie comprised 24 frames with a total cumulative dose of $\sim 65\,e^-/Å^2$.

**Image processing of ABC-Cdc42**. After initial screening of all micrographs, 2754 images were selected for further processing. Movie frames were aligned, dose-corrected and averaged using MotionCor2[49]. The integrated images were also used to determine the contrast transfer function parameters with CTER[50], implemented in the SPHIRE software package[43]. Initially, 2025 particles were manually picked and 2D class averages generated by Relion[51] were used as an autopicking template. In all, 99,980 particles were auto-picked from the images using the Relion 1.4 autopicker. Subsequently, reference-free 2D classification and cleaning of the dataset were performed with the iterative stable alignment and clustering approach ISAC[42] in SPHIRE. ISAC was executed with a pixel size of 7.2 Å/pixel on the particle level. The 'Beautifier' tool of SPHIRE was then applied to obtain refined and sharpened 2D class averages at the original pixel size, showing high-resolution features (Supplementary Fig. 5b). From the initial set of particles, the clean set used for 3D refinement contained 56,665 particles. We applied the previously obtained cryo-EM structure of ABC(WT) (EMDB-2551) as an initial model after scaling and filtering it to 12 Å resolution and performed 3D refinement in SPHIRE. The resolution of the final density was estimated to be 7.02/5.11 Å according to FSC 0.5/0.143 after applying a soft Gaussian mask. The B-factor was estimated to be $-246.4\,Å^2$. Local FSC calculation was performed using the Local Resolution tool in SPHIRE. (Supplementary Fig. 5e) and the electron density map was filtered according to its local resolution using the 3D Local Filter tool in SPHIRE. Cryo-EM data processing statistics are summarized in Supplementary Table 2.

**Reporting summary**. Further information on research design is available in the Nature Research Reporting Summary linked to this article.

## Data availability
The cryo-EM map of ABC-Cdc42 has been deposited in the Electron Microscopy Data Bank under accession number 10314. The coordinates of the crystal structures of TcB-TcC-Cdc42 and TcB-TcC-TEV have been deposited in the Protein Data Bank under accession numbers 6SUP and 6SUQ, respectively. The source data underlying Fig. 3a–c and Supplementary Figs 3a, b are provided as a Source Data file. Other data are available from the corresponding author upon request.

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

## Acknowledgements

We thank K. Vogel-Bachmayr for excellent technical support, and O. Hofnagel and D. Prumbaum for assistance with EM. We thank R. Tampé for a plasmid containing the ICP47 sequence and A. Ernst for a plasmid containing the RBD sequence. This work was supported by funding from the Max Planck Society (to S.R.) and the European Research Council under the European Union's Seventh Framework Programme (FP7/2007–2013 grant no. 615984, to S.R.).

## Author contributions

S.R. designed and supervised the project. D.R. designed proteins. D.R. performed the assembly and translocation experiments. D.R. and O.S. performed fluorescence spectroscopy. E.S. crystallized TcB-TcC-Cdc42 and TcB-TcC-TEV, and solved the crystal structures. D.R. processed and analyzed cryo-EM data of ABC-Cdc42. D.R., O.S., and S.R. wrote the manuscript.

## Competing interests

The authors declare no competing interests.
