## [Peer Review File · Nature Communications]

Reviewers' comments:

Reviewer #1 (Remarks to the Author):

The BC-subcomplexes of ABC toxin complexes (Tcs) are the best characterised family of RHS/YD-repeat containing proteins. The BC subcomplexes (and presumably other RHS/YD families) package their toxic C-terminal domain (the 'hypervariable region') inside their RHS/YD region, which forms a protective shell or cocoon, prior to forming the ABC holotoxin complex that is capable of translocating the toxin across the membrane of a target cell. The lack of sequence conservation amongst the peptides that are known or predicted to be packaged leads to the logical suggestion that RHS/YD proteins are 'sequence agnostic', and could therefore be engineered to deliver arbitrary proteins of choice into cells. In this paper, Roderer et al. set out to test this idea as a first step towards engineering ABC toxins as universal protein delivery machines.

The substantial amount of work presented in this paper marks a significant advance in our understanding of the characteristics of sequences that allow the formation of fully functional ABC holotoxin complexes able to translocate the contents of their RHS/YD cocoon. Overall, the data presented in the paper are compelling and will form the basis for future engineering of ABC toxins and other RHS/YD repeat proteins, and as such this paper is likely to be influential in the field. That said, I have a few comments, queries and suggestions for improvements:

Lines 183-189: The HVR45/HVR50/HVR128 nomenclature is confusing as written. The overall cloning strategy should be described more fully here to improve the clarity for the reader, rather than be just tucked in the methods.

Lines 234, 259 and also supplementary figure 3: The iLOV experiment is confusing, and the least convincing experiment presented. The fluorescence of iLOV is dependent on the folded protein binding to FMN (Chapman et al 2008), so in this case fluorescence can only function as a folding signal in the presence of FMN. Is it known whether the cofactor is incorporated within the cocoon? A priori this does not seem likely, given that the iLOV domain either never folds whilst being encapsulated, or unfolds during encapsulation - presumably the former. It is therefore hard to see what information the lack of fluorescence gives with regard to the folding status of the iLOV domain.

Line 257: Is TEV an exceptionally stable protein? The cited reference talks about the broad distribution of small beta-barrels, but not necessarily their exceptional stability, so the idea that TEV is 'too stable' seems somewhat speculative (see also 360). Is there any published data about TEV stability that can be cited to strengthen this argument?

Line 309: the use of the work 'occupied' here is confusing. Would it be better to say: "If the C-terminus that would normally be translocated first through TcA [16] is bound elsewhere, then why does Cdc42 form a holotoxin?"

Line 335: "the affinity drops tremendously" - the affinities have not been directly measured, but the ability to efficiently assemble holotoxins has been. I would suggest rewording this sentence to better match the data that has been presented.

Lines 860-881: Supplementary Figure 3 legend labelled incorrectly.

Line 932: As the Rmerge/Rmeasure values are high and the I/sigmaI rather low in the outer shell of data, it would be very useful to cite the CC1/2 values for the data, as described by Karplus and Diederichs, to justify the resolution cutoff for the data that is presented.

J. Shaun Lott, University of Auckland

Reviewer #2 (Remarks to the Author):

The manuscript by Roderer et al. presents a study to hijack the function of Tc-ABC toxins to translocate proteins other than the ~30 kDa cytotoxic component. This so-called cargo protein is linked to the Tc-C C-terminus and autoproteolytically cleaved prior to translocation into the target cell, where it performs its cytotoxic activity. The authors replaced the cargo by several other proteins with a size of 11-31 kDa and found that high-affinity binding of Tc-A to Tc-BC complex requires a minimum cargo size of 20-22 kDa and a maximum of ~32 kDa. Further, they conclude from their translocation experiments, that the cargo protein should have a pI of 8 or above and no extensive hydrophobic patches. Proteins that form highly stable structures within the Tc-BC (cocoon) also are not translocated by the Tc-ABC complex upon pore formation by pH shift from pH 8 to 11. To find out why the translocation is blocked, the authors crystallized the Tc-BC complex with bound CDC42 and solved the crystal structure of the Tc-CB-CDC42 at 2.0 Å using the wild-type Tc-CB structure as search model for molecular replacement. Analogous to the wild-type structure, the cargo was disordered and not visible in the crystal structure except a short helix stretch of the CDC42 C-terminus. Finally, the location of the short CDC42 helix was also verified by fitting the crystal structure of Tc-CB-CDC42 into the 5Å cryo-EM map of Tc-ABC-CDC42 complex obtained by single particle analysis.

All in all, the manuscript is dealing with a very interesting topic to hijack the function of Tc-ABC complex for translocation of proteins of interest into target cells. However, the prerequisites the cargo has to fulfill to be translocated by Tc-ABC, restrict the usability of this system as an "universal" protein translocation system considerably. In the presented experiments the cargos were released from the Tc-ABC complex and not translocated across a membrane. The question would be, whether translocation across the target membrane (or any other membrane) would further increase the prerequisites to the cargo even more? Another question is whether the cargo proteins are "translocated" to a functional form or whether they remain unfolded or aggregate as the experiment (Figure 3, Suppl. Figure 3) suggest and which is not discussed at all. Moreover, all "translocated" cargos are linked to the "HVR128" indicating that this region might play a more important role in the translocation. In the revised manuscript the issues above should be addressed and appropriately discussed.

Following some minor points:

Shorten and reduce the presented results within the introduction. E.g. almost the whole page six, lines 90-116 could be omitted from the introduction as the information is repetitive, e.g. to the results on page 8.

Lines 126-139: Is there a reason why the authors used the Tcd-B2-Tcc-C3-HVR fusion construct as base component for the cocoon scaffold and not the simple Tcc-C3-HVR?

Line 142: I guess the authors mean the replacement of the HVR from Tcd-B2-Tcc-C3-HVR and not of the whole Tcc-C3-HVR. Otherwise, if "Tcc-C3HVR" describes the cleaved region only, then the usage throughout the manuscript is not consistent, e.g. in Figure 1A and 1B only the "HVR" is translocated or replaced.

Figure 2C: While in the figure legend to Fig. 2C the authors write about HVR45-RBD-HVR50 and HVR130, in the figure itself HVR45-RBD and HVR128-RBD is shown. Moreover, the placement of the C panel is not optimal.

Figure 4C: The hydrophobic surface presentation shown here does not explain the interactions of the helix with the cocoon. Showing the interacting sidechains of TcCB in detail would help to understand the reason for the high-affinity binding and blocking by CDC42.

Suppl. Figure 2: The labeling of the A/B panels is wrong in the legend (D and E?).

Suppl. Figure 4: According to the results (lines 271-273) there are no differences in the structures shown in panel A. Therefore, showing three "identical" structures would not make so much sense.

We thank the reviewers for their helpful and constructive comments on our work and useful guidance in revising our paper. Below is a point-by-point response to all comments and a detailed description of all changes we have made to our manuscript after considering their suggestions.

Reviewers' comments:

Reviewer #1 (Remarks to the Author):

The BC-subcomplexes of ABC toxin complexes (Tcs) are the best characterised family of RHS/YD-repeat containing proteins. The BC subcomplexes (and presumably other RHS/YD families) package their toxic C-terminal domain (the 'hypervariable region') inside their RHS/YD region, which forms a protective shell or cocoon, prior to forming the ABC holotoxin complex that is capable of translocating the toxin across the membrane of a target cell. The lack of sequence conservation amongst the peptides that are known or predicted to be packaged leads to the logical suggestion that RHS/YD proteins are 'sequence agnostic', and could therefore be engineered to deliver arbitrary proteins of choice into cells. In this paper, Roderer et al. set out to test this idea as a first step towards engineering ABC toxins as universal protein delivery machines.

The substantial amount of work presented in this paper marks a significant advance in our understanding of the characteristics of sequences that allow the formation of fully functional ABC holotoxin complexes able to translocate the contents of their RHS/YD cocoon. Overall, the data presented in the paper are compelling and will form the basis for future engineering of ABC toxins and other RHS/YD repeat proteins, and as such this paper is likely to be influential in the field. That said, I have a few comments, queries and suggestions for improvements:

Lines 183-189: The HVR45/HVR50/HVR128 nomenclature is confusing as written. The overall cloning strategy should be described more fully here to improve the clarity for the reader, rather than be just tucked in the methods.

We agree with reviewer 1 that a more detailed description of the cargo constructs would be helpful here. In addition to the reference to the methods and Table 1, we added the following sentences (lines 168ff.):

“To achieve this, we introduced an EcoRI restriction site in frame after the first 138 or 384 base pairs (bp) of the sequence coding for the HVR, resulting in two possible N-terminal extensions of the cargo proteins with 45 or 128 residues from the original HVR, respectively. Additionally, we introduced a NotI restriction site 150 bp upstream of the stop codon, resulting in a C-terminal extension of the cargo with the C-terminal 50 residues of the HVR. Altogether, five different combinations of extensions of the cargos were possible: HVR45-cargo, HVR128-cargo, HVR45-cargo-HVR50, HVR128-cargo-HVR50 and cargo-HVR50.”

Lines 234, 259 and also supplementary figure 3: The iLOV experiment is confusing, and the least convincing experiment presented. The fluorescence of iLOV is dependent on the folded protein binding to FMN (Chapman et al 2008), so in this case fluorescence can only function as a folding signal in the presence of FMN. Is it known whether the cofactor is incorporated

within the cocoon? A priori this does not seem likely, given that the iLOV domain either never folds whilst being encapsulated, or unfolds during encapsulation - presumably the former. It is therefore hard to see what information the lack of fluorescence gives with regard to the folding status of the iLOV domain.

The iLOV fluorescence experiment has been indeed presented in a confusing way. We have shown fluorescence spectra of iLOV, TcB-TcC-iLOV and TcB-TcC-HVR128-iLOV, but we only refer to HVR128-iLOV in the manuscript. We have rephrased this paragraph and the figure legend in the revised manuscript.

The experiment in Supplementary Figure 3d shows that TcB-TcC-iLOV is fluorescent with ~10% measured fluorescence intensity in comparison to purified iLOV at an identical concentration, whereas TcB-TcC-HVR128-iLOV is non-fluorescent. We therefore conclude that FMN is present inside the cocoon, otherwise TcB-TcC-iLOV could not be fluorescent. The fact that TcB-TcC-HVR128-iLOV is non-fluorescent shows that the additional 128 residues of the HVR prevent folding inside the cocoon. We have rephrased the paragraph and the figure legend (Supplementary Fig. 3) in the revised manuscript.

Line 257: Is TEV an exceptionally stable protein? The cited reference talks about the broad distribution of small beta-barrels, but not necessarily their exceptional stability, so the idea that TEV is 'too stable' seems somewhat speculative (see also 360). Is there any published data about TEV stability that can be cited to strengthen this argument?

TEV protease wild type, which we used in this study, unfolds at ~1.3 M GdmCl or 52 °C (Cabrita et al., Protein Sci, 2007), which does not make it an extraordinarily stable protein. However, TEV contains small β -barrels, which are typically stable folding units (Youkharibache et al, Structure, 2019). Therefore, we believe that these domains are quite stable and more rigid than wildtype HVR.

To prove this, we have now determined the structure of TcB-TcC-TEV by X-ray crystallography to a resolution of 3.7 Å (see Supplementary Table 1). While the cocoon is almost identical to that of TcB-TcC(WT) (pdb ID 4O9X) with a C_{α} RMSD of 0.552 Å², the electron density inside appears already at a higher sigma level, indicating that the encapsulated TEV has a lower degree of flexibility and disorder than the HVR. However, the limited resolution of the crystal structure did not allow us to build a partial model of TEV into the density. We therefore show a comparison of both crystal structures with the density inside at identical binarization thresholds (Supplementary Figure 3c).

We have changed the corresponding section of the Methods part to include crystallization of TcB-TcC-TEV (lines 595ff.), and we describe now the comparison of the crystal structures of TcB-TcC-TEV in the results section (lines 263ff. in the revised manuscript).

Line 309: the use of the work 'occupied' here is confusing. Would it be better to say: "If the C-terminus that would normally be translocated first through TcA [16] is bound elsewhere, then why does Cdc42 form a holotoxin?"

We thank reviewer 1 for the suggestion and rewrote the sentence accordingly.

Line 335: “the affinity drops tremendously” - the affinities have not been directly measured, but the ability to efficiently assemble holotoxins has been. I would suggest rewording this sentence to better match the data that has been presented.

We agree with reviewer 1 and reworded the sentence to:

“Importantly, we discovered that the lower size limit of the cargo is in the 20 – 22 kDa range, and the ability to form the ABC holotoxin is drastically reduced if the cargo size is below this limit (Fig. 5, Table 1).”

Lines 860-881: Supplementary Figure 3 legend labelled incorrectly.

We apologize for the error and corrected the labeling of the figure legend.

Line 932: As the Rmerge/Rmeasure values are high and the I/sigmaI rather low in the outer shell of data, it would be very useful to cite the CC1/2 values for the data, as described by Karplus and Diederichs, to justify the resolution cutoff for the data that is presented.

We thank reviewer 1 for pointing this out and added the CC1/2 value (0.707 and 0.688 for the highest resolution shells of TcB-TcC-TEV and TcB-TcC-Cdc42, respectively) to Supplementary Table 1.

J. Shaun Lott, University of Auckland

We thank J. Shaun Lott for taking the time to review our manuscript and his constructive suggestions. We appreciate that he revealed his identity.

Reviewer #2 (Remarks to the Author):

The manuscript by Roderer et al. presents a study to hijack the function of Tc-ABC toxins to translocate proteins other than the ~30 kDa cytotoxic component. This so-called cargo protein is linked to the Tc-C C-terminus and autoproteolytically cleaved prior to translocation into the target cell, where it performs its cytotoxic activity. The authors replaced the cargo by several other proteins with a size of 11-31 kDa and found that high-affinity binding of Tc-A to Tc-BC complex requires a minimum cargo size of 20-22 kDa and a maximum of ~32 kDa. Further, they conclude from their translocation experiments, that the cargo protein should have a pI of 8 or above and no extensive hydrophobic patches. Proteins that form highly stable structures within the Tc-BC (cocoon) also are not translocated by the Tc-ABC complex upon pore formation by pH shift from pH 8 to 11. To find out why the translocation is blocked, the authors crystallized the Tc-BC complex with bound CDC42 and solved the crystal structure of the Tc-CB-CDC42 at 2.0 Å using the wild-type Tc-CB structure as search model for molecular replacement. Analogous to the wild-type structure, the cargo was disordered and not visible in the crystal structure except a short helix stretch of the CDC42 C-terminus. Finally, the location of the short CDC42 helix was also verified by fitting the crystal structure of Tc-CB-CDC42 into the 5Å cryo-EM map of Tc-ABC-CDC42 complex obtained by single particle analysis.

All in all, the manuscript is dealing with a very interesting topic to hijack the function of Tc-ABC complex for translocation of proteins of interest into target cells. However, the prerequisites the cargo has to fulfill to be translocated by Tc-ABC, restrict the usability of this system as an “universal” protein translocation system considerably. In the presented experiments the cargos were released from the Tc-ABC complex and not translocated across a membrane. The question would be, whether translocation across the target membrane (or any other membrane) would further increase the prerequisites to the cargo even more?

There is no indication that the transport across a membrane imposes any additional requirements for cargo translocation in case of Tc toxins. The HVR is also translocated and released from wild type ABC toxin that is embedded in lipid nanodiscs without the need for a driving force such as a proton gradient or chaperones (Roderer, Hofnagel *et al.*, biorxiv 2019, under revision), although a facilitation of translocation by the latter two factors cannot be excluded in the context of a cellular environment.

The A-component of the toxin can insert into the membrane and form a channel even in the absence of the B-C component(s) and its cargo, as seen by cell toxicity assays (Waterfield *et al.*, Cell Microbiol., 2005) and single channel conductivity experiments (Gatsogiannis *et al.*, Nature, 2013). Additionally, when inserted into a membrane, no lipids or cofactors are present within the channel (Gatsogiannis *et al.*, NSMB, 2016, Roderer, Hofnagel *et al.*, biorxiv 2019, under revision). This means that cargo proteins can move unhindered through the transmembrane channel provided their pI is basic enough to do so, a situation mimicked by our translocation assay. We added a paragraph to the discussion (lines 370ff) to address these points.

Another question is whether the cargo proteins are “translocated” to a functional form or whether they remain unfolded or aggregate as the experiment (Figure 3, Suppl. Figure 3) suggest and which is not discussed at all.

We thank reviewer 2 for mentioning this important aspect. We already show for TccC5HVR, which is a heterogeneous cargo, that it is translocated across the cell membrane and functional *in vivo*, as evidenced by the intoxication shown in Figure 1c.

In order to prove that proteins also fold and are functional after translocation *in vitro*, we performed additional experiments that are described here: lines 216ff, 492ff., 565ff, Fig. 3b. We reconstituted the holotoxin in nanodiscs and isolated the translocated ADP-ribosyltransferase and compared its activity with recombinantly purified TccC3HVR. At concentrations of 50 to 500 pM of TccC3HVR, the translocated TccC3HVR successfully ADP-ribosylates F-actin similarly to the recombinantly purified enzyme, as judged by immunodetection of ADP-ribosylated actin. This demonstrates that most of the translocated ADP-ribosyltransferases were folded to their active form after translocation. Importantly, folding and activity of the ADP-ribosyltransferase do not require additional cofactors such as chaperones and we believe that this is also the case for other translocated cargos.

Moreover, all “translocated” cargos are linked to the “HVR128” indicating that this region might play a more important role in the translocation. In the revised manuscript the issues above should be addressed and appropriately discussed.

TccC5HVR is an example of a heterogeneous cargo which is not linked to the HVR128. It is both translocated and functional after translocation across the cell membrane, as evidenced by the intoxication shown in Figure 1c. Importantly, there is almost no similarity between the first 128 amino acids of TccC5HVR and TccC3HVR (see below and also Supplementary Figure 1b in the revised manuscript). This indicates that the sequence of the first 128 amino acids of the latter (HVR128) are not a prerequisite for translocation, but rather its length.

We addressed the difference between the HVRs also in the results section (lines 124ff. in the revised manuscript):

“Importantly, the two different HVRs do not show any pronounced sections of sequence identity (Supplementary Figure 1b), indicating that there is no conserved motif in Tcc HVRs that is a general prerequisite for successful translocation.”

```

TccC3HVR   1  MPTIAERIAAALKKNKVTDSAPSPANATNVAINLRPVPAPKPSLPKASTSSOPTTH-----P--LGAANKKPTTSGSSIVA
TccC5HVR   1  MPEFRTEEAIIKQGSFTGMEEAVYKK-----MAKQTEKRQRATAAQIEQE-AHESLTNNPSSDISPKNYTIDSSQIN

TccC3HVR   74  PLSFVGNKSTSEISLPEAQSSSSSTTTNIOKKSFTLYRADNRSFEEMQSKFPEGFKAWTPLDTKMARQFASIFTCQKD
TccC5HVR   74  A-AIRENRITPAV---ESLDATLSSLQDROMRVTYRVMTYVDNSIP-----SPWISPOEGNSINVCDIV

TccC3HVR   154 TSNLPKETVKNISTGAKPKLRDLSNYIKYTKDKSTVWVSTAINTEAGGSSGAPLHKIDMDLYEFAID-----GQK
TccC5HVR   134 SDN-----ALSTSAHRGFLNEVHKKETSETRIVKMAFLTNAQVNVSLASLYNNAGEEQVFKMDLNSRKSLEK

TccC3HVR   226 LNPLPEERTKNMVPSLLLDTPQIETSSITAINHGP-----VNDAEISFLTIPLKNVKPKHR
TccC5HVR   204 LKLRVSEPPQSGQAEILLPRETOFEV---VSLRHQGRDITYVLLQDINQSAATHRNVRNTYTGNFKSSAN-----

```

Following some minor points:

Shorten and reduce the presented results within the introduction. E.g. almost the whole page six, lines 90-116 could be omitted from the introduction as the information is repetitive, e.g. to the results on page 8.

We thank reviewer 2 for this suggestion and shortened the last two paragraphs of the introduction. We placed the detailed description of the cargo proteins and the respective citations into the results section instead (lines 135ff in the revised manuscript).

Lines 126-139: Is there a reason why the authors used the Tcd-B2-Tcc-C3-HVR fusion construct as base component for the cocoon scaffold and not the simple Tcc-C3-HVR?

The cocoon is formed by TcdB2 and TccC3 together, TccC3 alone forms only ~30% of the cocoon and does not form a functional unit. In contrast to the individual proteins TcdB2 and TccC3, the fusion protein TcdB2-TccC3, which we also used for all our previous studies, can be expressed and purified in sufficient amounts to allow subsequent biochemical and structural characterization. With the experiment shown in Figure 1c, the replacement of TccC3HVR by TccC5HVR in the context of TcdB2-TccC3 and the obtained cell toxicity of the chimeric toxin, we showed that the TcdB2-TccC3 fusion is in general suitable as a cocoon scaffold.

We agree with reviewer 2 that mentioning the fact that TcdB2-TccC3 is a fusion protein, which is stated in the Methods section, here in this paragraph of the Results section is confusing and we therefore reworded the sentence to:

“This experiment shows that the cocoon formed by TcdB2-TccC3 is capable of also encapsulating and translocating other HVRs, such as TccC5HVR.”

Line 142: I guess the authors mean the replacement of the HVR from Tcd-B2-Tcc-C3-HVR and not of the whole Tcc-C3-HVR. Otherwise, if “Tcc-C3HVR” describes the cleaved region only, then the usage throughout the manuscript is not consistent, e.g. in Figure 1A and 1B only the “HVR” is translocated or replaced.

We apologize for the non-consistent usage of HVR and TccC3HVR and changed the sentence to:

“Our next step was to replace the HVR of TccC3 with unrelated heterologous proteins and test the capability of the holotoxin to translocate these.”

Figure 2C: While in the figure legend to Fig. 2C the authors write about HVR45-RBD-HVR50 and HVR130, in the figure itself HVR45-RBD and HVR128-RBD is shown. Moreover, the placement of the C panel is not optimal.

We apologize for the error and corrected the figure legend, as HVR45-RBD and HVR128-RBD is correct. We also increased the gap between panels A and C to avoid the overlay of both figure descriptions.

Figure 4C: The hydrophobic surface presentation shown here does not explain the interactions of the helix with the cocoon. Showing the interacting sidechains of TcCB in detail would help to understand the reason for the high-affinity binding and blocking by CDC42.

We agree with reviewer 2 and added an additional representation, showing the side chains of the region of TcdB2 and the helix of Cdc42 (Figure 4d). However, in our opinion the hydrophobic surface representation of the binding site is also helpful to understand the nature of interaction, we therefore have also kept this representation in Figure 4.

We also refer to the side chain contacts and Figure 4d in the results section (lines 297ff. in the revised manuscript):

“The side chains of F169, I173, L177 and P179 are rigidly oriented towards the hydrophobic cleft, where they are in close contact with L39, L41, P42, L366, L368, M702, N704, V708, H710, L1203 and F1349 of the cocoon (Fig. 4d). In contrast, D170, E171 and E178 face the TcB-TcC lumen with more degrees of freedom.”

Suppl. Figure 2: The labeling of the A/B panels is wrong in the legend (D and E?).

We apologize for the error and corrected the labeling of the figure legend.

Suppl. Figure 4: According to the results (lines 271-273) there are no differences in the structures shown in panel A. Therefore, showing three “identical” structures would not make so much sense.

We agree with reviewer 2 that the three cocoons show an identical shape, and that pointing out this fact in a Supplementary Figure panel is obsolete. We therefore removed panel A from Supplementary Figure 4 and wrote the backbone RMSD values in the text in the results section in the revised manuscript (lines 285ff):

“The overall shape of the TcB-TcC-Cdc42 cocoon is identical to wild-type (WT) and empty TcB-TcC (Meusch *et al.*, 2014; Gatsogiannis *et al.*, 2018), with C_{α} -RMSD values of 0.424 Å² and 0.414 Å² between TcB-TcC-Cdc42/TcB-TcC WT and TcB-TcC-Cdc42/TcB-TcC empty, respectively, indicating that a different cargo does not influence the RHS repeat structure of the cocoon.”

We thank reviewer 2 for taking time to review our manuscript and the constructive suggestions.

REVIEWERS' COMMENTS:

Reviewer #1 (Remarks to the Author):

The authors have done a good job in addressing my previous queries and in clarifying the manuscript in places where the original phrasing was unclear. The additional description of how the HVR constructs were produced will be very helpful to the reader.

The clarification around the stability of the small β -barrels of TEV is helpful, and the new structure presented is consistent with, if not proof of, small regions of the TEV structure being able to fold in the lumen of the cocoon.

I still find the iLOV experiment the most difficult to rationalise. Presumably what is happening, is that in the case of the TcB-TcC-iLOV construct, a small proportion (i.e. 10%) of the encapsulated iLOV molecules are able to fold, and has access to FMN, within the cocoon, within the bacterial cell. This would be consistent with the TEV result, and also with the presumed ability of small molecule cofactors to be able to access the interior of the cocoon. However, when the cocoon is "fuller", in the case of the TcB-TcC-HVR128-iLOV, there is no evidence of even inefficient folding. Do I have this reasoning correct? If so, I think it would help the reader understand the experiment if this was more explicitly stated in the text of the paper.

I'd be more than happy to see this paper published, with this minor additional clarification in place.

Reviewer #2 (Remarks to the Author):

In the revised version of the paper, the authors responded to all my concerns and considerably improved the manuscript so that it is ready to be published.

Best regards

REVIEWERS' COMMENTS:

Reviewer #1 (Remarks to the Author):

The authors have done a good job in addressing my previous queries and in clarifying the manuscript in places where the original phrasing was unclear. The additional description of how the HVR constructs were produced will be very helpful to the reader.

The clarification around the stability of the small β -barrels of TEV is helpful, and the new structure presented is consistent with, if not proof of, small regions of the TEV structure being able to fold in the lumen of the cocoon.

I still find the iLOV experiment the most difficult to rationalise. Presumably what is happening, is that in the case of the TcB-TcC-iLOV construct, a small proportion (i.e. 10%) of the encapsulated iLOV molecules are able to fold, and has access to FMN, within the cocoon, within the bacterial cell. This would be consistent with the TEV result, and also with the presumed ability of small molecule cofactors to be able to access the interior of the cocoon. However, when the cocoon is "fuller", in the case of the TcB-TcC-HVR128-iLOV, there is no evidence of even inefficient folding. Do I have this reasoning correct? If so, I think it would help the reader understand the experiment if this was more explicitly stated in the text of the paper.

We observed iLOV fluorescence in the TcB-TcC cocoon when only iLOV alone was incorporated, but not when iLOV was fused to HVR128, therefore the reasoning is correct. This is already described in the legend of Supplementary Figure 3. However, we agree that a description of these experimental results in the main text would be helpful and extended the section in the manuscript accordingly (lines 242 ff.):

“We conclude that extending the N-terminus by 128 additional residues prevents the folding of HVR128-iLOV inside the cocoon, while the fluorescence of TcB-TcC-iLOV shows that iLOV without the extension is able to fold inside the cocoon and has access to the cofactor FMN.”

I'd be more than happy to see this paper published, with this minor additional clarification in place.

Reviewer #2 (Remarks to the Author):

In the revised version of the paper, the authors responded to all my concerns and considerably improved the manuscript so that it is ready to be published.

Best regards

We again thank both reviewers for their constructive comments that helped us to improve the manuscript.